# Ontology for Informatics Research Artifacts*

Viet Bach Nguyen(iD) ✉ and Vojtěch Svátek(iD)

Department of Information and Knowledge Engineering
Prague University of Economics and Business, Czech Republic
{nguv03,svatek}@vse.cz

**Abstract.** The IRAO ontology, as a new contribution to the network of ontologies for the scholarly domain, aims to model the most tangible aspect of research in computing disciplines – the research artifacts. It consists of parts focusing on the concepts of researcher, research artifact classification, research artifact meta information, relationships between artifacts, and research artifact quality evaluation benchmarks that are used to express the quality and maturity of each research artifact. We describe the ontology design requirements using competency questions and the evaluation of the ontology by the same questions that helped in defining the concept domain coverage.

**Keywords:** competency question, ontology building, ontology competency, ontology design, research artifact, research output

## 1 Introduction

Engineering disciplines such as informatics[1] tend to deliver, aside from the contribution to scientific knowledge by testing hypotheses and building theories, also compact, 'tangible' outputs, namely *artifacts*. Most typical examples are software prototypes, benchmark (or other) datasets, ontologies, as well as methodologies or managerial frameworks produced by Information Systems researchers. Notably, several computer science conferences (ISWC, ESWC, CIKM, BPM, SIGIR) and some journals have recently started to use the term 'resource paper' to denote a paper describing a reusable artifact aiming to serve the research (or, sometimes even practitioner) community; specific subclasses are then those of, e.g., 'software paper', 'dataset paper' or 'ontology paper'. Given this central role of artifacts in informatics research, it is rather surprising that no ontology has so far paid particular attention to this topic, by our recent survey [2].

A research artifact is a tangible research output of a research project. This artifact is then circulated, shared, published, further developed, and may be

---

* Supported by IGA VŠE project № 56/2021

[1] We use this concise term as largely interchangeable with 'computing disciplines', as discussed, e.g., in the new proposal for ACM/IEEE Computing Curricula, see https://cc2020.nsparc.msstate.edu/wp-content/uploads/2020/11/Computing-Curricula-Report.pdf.

reused to advance other research projects or applied in real-world scenarios. The motivation for carefully modeling informatics artifacts as a compact domain (rather than just specific kinds of artifacts in isolation) is manifold:

- The occurrence of different kinds of artifacts in research publications can be traced over time for particular sub-disciplines or venues, thus providing a broad picture of trends in informatics research.
- Networks of complementary or competitive artifacts (such as software tools being developed using a given methodology and applied on specific datasets backed on particular ontologies) can be connected together, thus allowing researchers to rapidly navigate from one to another and finding a reuse target (and even associated publications) more easily than by keyword search.
- Similarly, industrial companies can retrieve artifacts that they might consider transforming into deployed products.

In this poster paper, we introduce an ontology solution for the representation and management of informatics research artifacts. The goal of this ontology is to capture knowledge about research artifacts in the researcher environment and provide a reference model for the academic domain. This ontology is evaluated using competency questions that define the use cases for its functionalities.

Research/academia is gradually becoming a mainstream target domain for ontology-based applications. Recently we have published a comprehensive survey and analysis of academic and research-related ontologies [2]. We have retrieved and analyzed 43 ontologies and created a holistic model mapping for their coverage based on competency questions that focus on the academic domain from the perspective of a (primarily, senior) researcher's information needs. Among the reviewed resources, no ontologies focus on the description nor evaluation of research outputs or artifacts. There are, however: ontologies that include some general terms like *Resource*, in e.g., CCSO or DataCite;[2] ontologies that cover the project aspect of research with terms like *Project*, *License* or *Repository*, e.g., DOAP or SWRC;[3] ontologies that cover the publishing part of research with terms like *Deliverable* or *Output*, in e.g., VIVO or FRAPO.[4]

We have concluded that the artifact aspect of research needs an overarching formal conceptualization, which we identify as one of the missing features (gaps) in the existing ontology eco-system of the academic/researcher domain. By our survey, none of the current ontologies fully cover the requirement specification of our use cases, including the tangibility and quality assessment aspects. Another important requirement is the classification of research artifacts and the relationship between their types, which can be used to capture the interdependencies between artifacts within research projects.

---

[2] http://xworks.gr/ontologies/ccso, http://purl.org/spar/datacite

[3] http://usefulinc.com/ns/doap, http://swrc.ontoware.org/ontology

[4] http://vivoweb.org/ontology/core, http://purl.org/cerif/frapo

## 2 Ontology design and competency questions

Conforming to the NeOn ontology engineering methodology, we list out a set of competency questions (CQ) as part of our requirement specification document (ORSD) [3] to elicit relevant concepts, e.g., *CQ10 What type of artifact is it?* Both the CQs and the ORSD can be found in our GitHub repository.[5]

We first look up the terms 'research output' and 'research artifact' as defined on the websites of several academic institutions/universities (see the ORSD). From the definitions we collect the high-level entities such as *Researcher*, *Research Project* or *Research Artifact*. The *Informatics Research Artifact* as the direct output of a *Research Project* is the focal point of our ontology. To organize our competency questions, we divide the features of our ontology to four feature groups: for the research artifact itself, its development/readiness, its publishing/visibility and its quality.

Based on the gathered definitions of research output and competency questions, the ontology model should feature the following concepts:

- basic information about research artifacts, required for representing the artifact data gathered from repositories of theses, publications, software data repositories – incl. authorship, publication date, research field, topic, identifiers, etc.,
- types of research artifacts in terms of what they are useful for and how to use them,
- their development status, e.g., alpha, beta, release, or numbered version,
- their quality attributes, such as accessibility, use of an open standard, accessibility, or design principles,
- relationships between different types of artifacts, e.g., a dataset is described by a data model, a software uses a framework, etc.

## 3 Ontology construction and evaluation

Informatics Research Artifact Ontology (IRAO) was implemented in OWL using the Protégé editor. We also used OnToology [1] to automatically build the ontology using recommended metadata properties for self-documentation. The code and diagrams of IRAO can be found in our GitHub repository. The diagram in Fig. 1 shows the main entities of the ontology. The ontology documentation is available at `https://w3id.org/def/InformaticsResearchArtifactsOntology`.

IRAO consists of four parts that model the mentioned features. The artifact classification part lists out possible types of artifacts, e.g., *Dataset*, *Framework*, *Vocabulary*, and *Methodology*. All these types are defined as subclasses of the main concept. The meta information part includes relationships such as *hasAuthor*, *hasPublication*, *hasDomain* or *hasField*, having the range of *Researcher*, *Publication*, *Domain* and *Field*, respectively. The property *hasDevelopmentStatus* points to information about the maturity of the artifact. Properties

---

[5] `https://github.com/nvbach91/informatics-research-artifacts-ontology`

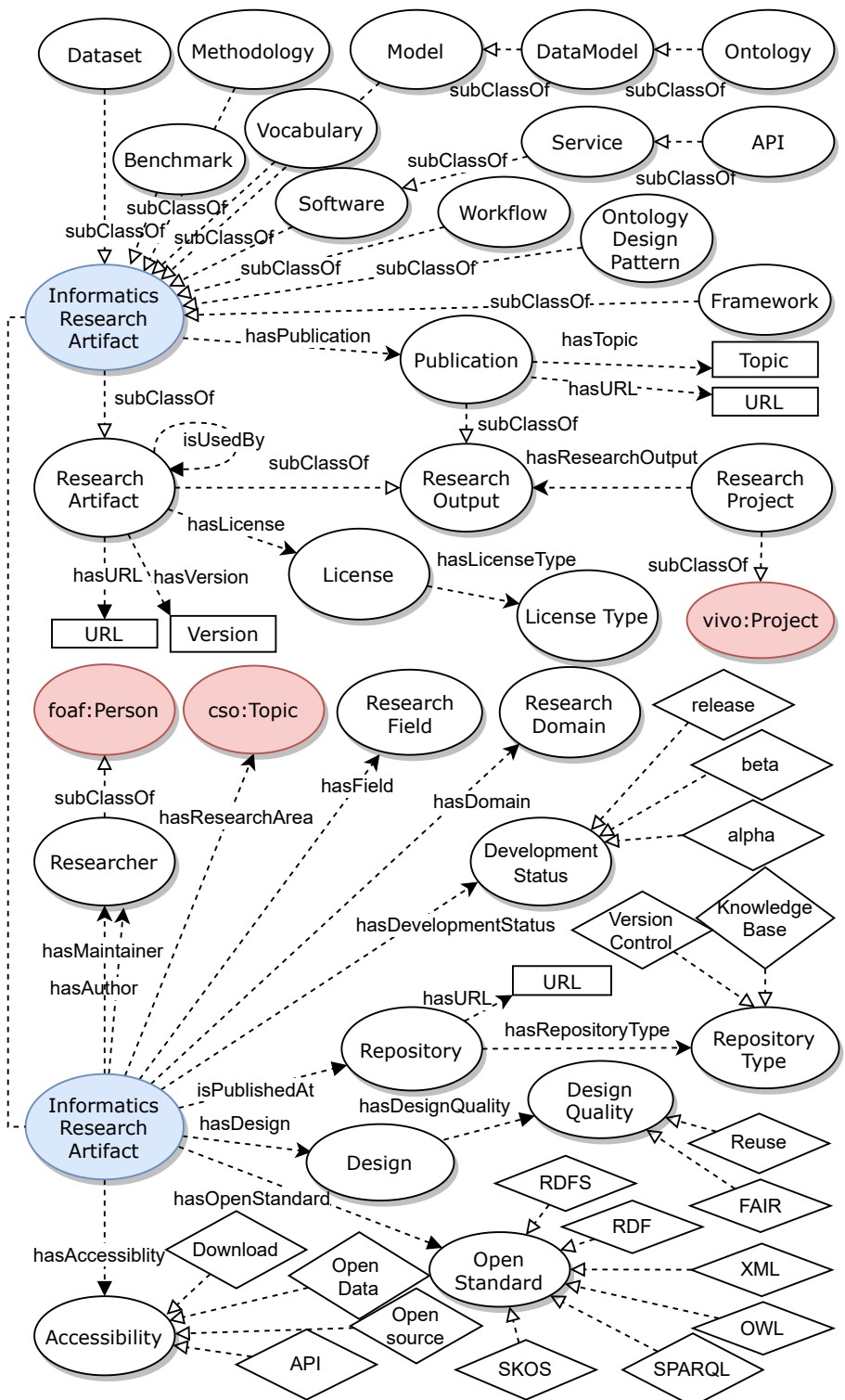

**Fig. 1.** Informatics Research Artifact Ontology main diagram

*hasAccessibility*, *hasDesignQuality*, *isPublishedAt* and *hasStandard* are used to provide the artifact with classifying tags such as what types of standard was used to create the artifact and in what way is the artifact made available. The last part of our ontology deals with the relationships between different types of artifacts to describe situations such as when a research project can produce multiple types of artifacts in a way that some types of artifact can precede or follow other types of artifact. The diagram for these relationships is available on GitHub.

To ensure that the definitions correctly implement the ontology requirements and competency questions, we created several SPARQL queries to answer all competency questions. For example, to answer *CQ02 Who is the artifact's creator?* and *CQ11 Where was the artifact created?* we can use the following query:

```
SELECT * WHERE { ?artifact a irao:ResearchArtifact .
                ?artifact irao:hasAuthor ?author .
                ?author irao:hasAffiliation ?institution1 . }
```

**Listing 1.1.** Query for artifact author and affiliation

We also provide several examples of manually populated instance data for the use cases of conference resource track contributions and informatics dissertations in our GitHub repository.

## 4 Conclusion

The ontology described in this paper is envisioned to capture and manage informatics research artifacts. It can be used as a data model in a knowledge graph to track research artifacts along with their characteristics and meta-information, including relationship with other artifacts. It provides quality criteria for evaluating and recommending artifacts for researchers, reviewers and potential industrial adopters. The evaluation of the ontology was carried out by translating its competency questions to SPARQL queries, and examples from two different use cases were formulated to demonstrate its wider scope.

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
