# OpenReview forum: "Ontology for Informatics Research Artifacts"
_eswc-conferences.org/ESWC/2021/Conference/Poster_and_Demo_Track — ESWC2021 P&D_

### Official Review · AnonReviewer2 · 2021-04-07
**Ontology in a domain that lacks ontologies**

**Rating:** 8
**Confidence:** 3

**Review:**

This poster paper presents the IRAO ontology (Informatics Research Artifacts Ontology) that aims to model research artifacts from computer science. For instance, it contains concepts like datasets, software, ontologies, API, etc., in other words, outputs of a research project. The authors published a survey last year in EKAW about Ontologies Supporting Research-Related Information, and concluded that there is no ontology dealing with this topic. The motivations of this ontology are to see trends in informatics research, and to facilitate reuse and deployment by linking complementary (or competitive) artifacts.

The methodology to build the result ontology was to list competency questions, like “what type of artifact is it?”. They also extracted high-level entities from online definitions of research output and research artifact. Therefore, they defined concepts and relationships between them. The ontology is available. The authors also provide instance examples, and SPARQL queries to answer competency questions.

Pros
-	Well written
-	A lot of information available online
-	Original: no existing ontology

Cons
-	The reader may have difficulty visualizing the ontology if he/she does not go to the github website. In case the paper gets accepted I would suggest the authors to put at least the picture of the main diagram (available on their github) in the extra page.


In conclusion, the paper presents a novel ontology describing the domain of research artifacts in informatics. It could be used in the future for several purposes: evaluation of trends, reusability and recommendation of artifacts, deployment in the industry. I accept this paper.

**Anonymity:**

Yes, I would like my review to remain anonymous.

---

### Official Review · AnonReviewer4 · 2021-04-13
**The validation is not well explained, but the ontology meets the requirements**

**Rating:** 6
**Confidence:** 4

**Review:**

The authors' previous studio revealed that current ontologies for research domains are not enough for modeling artifacts (e.g., benchmark datasets, ontologies, software, etc.). With this motivation, this paper proposes the ontology IRAO for modeling artifacts and their researcher environments. The authors created a set of 16 competency questions (available only in GitHub) that need to be answered using the data modeled by the ontology shown next,

- CQ01. What is the artifact's name?
- CQ02. Who is the artifact's creator?
- CQ03. What is the artifact's purpose?
- CQ04. What technology is used to create the artifact?
- CQ05. What maturity stage is the artifact in?
- CQ06. Where is the artifact published?
- CQ07. How is the artifact's licensed?
- CQ08. What is the impact of the artifact?
- CQ09. What field is the artifact targeting?
- CQ10. What type of artifact is it?
- CQ11. Where was this artifact created?
- CQ12. How is the artifact made available?
- CQ13. What is the artifact's design qualities?
- CQ14. How is the artifact made findable?
- CQ15. What other artifacts does this artifact make use of?
- CQ16. What is the relationship between specific artifacts?

The ontology is validated by the construction of SPARQL queries that could answer each of these competency questions.

With the increase of the number of publication of ontologies, some works (e.g., Baker et al.) highlight rules to accomplish:
  - the principle of "safety through redundancy" which advocates for mirroring information online.
  - submit the ontology to ecosystems as Linked Open Vocabularies (https://lov.linkeddata.es/dataset/lov/) to gain visibility and facilitate its reuse.
  - provide labels and definitions for each class in natural language to improve human readability
  - specify the maintainer (maybe using DOAP ontology)
  - others

Maybe this paper fulfills many of them, and it could be interesting know about that. Next, some strengths and weaknesses.

Strengths:
- The proposed ontology is new and relevant for the Semantic Web community.
- The ontology is available online, as well as examples and related documentation.
- This ontology has external links to foaf, vivo, cso

Weaknesses:
- The paper is not easy to read.
- The competency questions are a key component in this research that spots its need. However, there is no much information about them. Why those specific questions? Which were the criteria to create them? Is this set of questions extendable?


Reference
---------
 T. Baker, P. Vandenbussche, B. Vatant, Requirements for vocabulary preservation and governance, Library Hi Tech 31 (2013) 657–668.


**Anonymity:**

Yes, I would like my review to remain anonymous.

---

### Official Review · AnonReviewer1 · 2021-04-15
**Review for Ontology for Informatics Research Artifacts**

**Rating:** 4
**Confidence:** 4

**Review:**

This paper aims at representing an ontology named IRAO, i.e., Informatics Research Artifact Ontology, for managing artifacts in informatics research. The described OWL ontology is able to capture information about artifacts, their development status, publication status, as well as their quality. Unlike previously studied ontologies focusing on the academic domain, IRAO classifies the artifacts and models the relations between them. A nicely structured documentation page is also provided.
A more thorough evaluation would have been nice to compare better with existing ontologies in the domain. Also, it has not been mentioned if the same ontology can be used for other fields than Informatics or not.
In general, the ontology is novel itself, however the approach doesn't bring much novelty. The paper's writing in its current version is not really favorable and can be improved.

---- Some Minor remarks----
Some sentences are too long, making them hard to follow.

Introduction, page1, first paragraph: also called artifacts --> namely artifacts. But also --> as well as. recently started --> have recently started. Researcher community --> research community.
Introduction, page2, second paragraph: such as particular software tools being developed using a particular methodology and applied on particular datasets backed on particular ontologies) --> replace the word particular that has been used 4 times with: given, specific, different, etc. sown --> connected. ,industrial companies can this way retrieve artifacts that they might consider transforming into deployed products. --> This way, companies can retrieve artifacts they aim to transform into actual products.
 Introduction, page 2, last paragraph : the sentence is too long and hard to read and should be rewritten. The tense of the paper can be present rather than past.

**Anonymity:**

Yes, I would like my review to remain anonymous.

---

### Decision · Program_Chairs · 2021-04-19

Accept